# Hydrodynamic Response of Ocean-Towed Cable-Array System under Different Munk Moment Coefficients

**Dapeng Zhang [1], Bowen Zhao [2],\* and Keqiang Zhu [3]**

1   College of Ocean Engineering, Guangdong Ocean University, Zhanjiang 524088, China; zhangdapeng@gdou.edu.cn
2   Ocean College, Zhejiang University, Zhoushan 316000, China
3   Faculty of Maritime and Transportation, Ningbo University, Ningbo 315211, China; zhukeqiang@nbu.edu.cn
\*   Correspondence: zju_zhbw@zju.edu.cn

**Abstract:** The ocean towing system plays an important role in the ocean development process. The motion of a towed body is closely coupled with the motion of a towing cable. In this paper, the lumped mass method is used to discrete a towing cable into a lumped mass model. At the same time, on the basis of some assumptions, the relationship between the expression of Munk moments in the classical towed body kinematics and the expression of the Munk moments in the hydrodynamic analysis software OrcaFlex is established. Based on the above assumptions and the derivation, combined with the specific parameters of a certain sea state and a certain towing system, the dynamic simulation of the towing system is made by OrcaFlex. By changing the different Munk moment coefficients, the real-time response of the cable tension and the towed underwater body under different Munk moments is achieved. The effects of different Munk moment coefficients on the change of the tension are obtained; the six degrees of freedom of the towed body under the action of different Munk moment coefficients are shown. To obtain the spectral density of the six degrees of freedom of the towed body under the action of different Munk moment coefficients, Fast Fourier Transform is performed on the calculated results of the towed body in the time domain. The results provide a theoretical basis for the optimal design of a cable and towed body.

**Keywords:** towing system; towed body; OrcaFlex; Munk moment coefficients

## 1. Introduction

The survival and development of human beings are inseparable from sustainability. The second half of the 20th century to the present day, represents a golden age to actively develop ocean resources. In order to obtain various industrial raw materials in the ocean and realize the sustainable development of marine resources, an ocean towing system has been widely used in many fields, such as ocean monitoring, military detection, seabed mapping and naval defense. Among them, an automatic towing instrument for detecting ocean mineral resources, a towing sonar for searching and detecting offshore seabed, a trawl system for capturing deep-sea fish resources and a towing system for laying underwater cables can greatly improve sustainability.

In many types of applications, the systems include a towing ship, towing cable and towed body. Normally, the towed body is equipped with many types of ocean detection instruments. The operation of instruments in a towing system requires the towed body to be stable during a towing operation. However, under real ocean environments, the unsteady motion of a towing ship is transmitted down the cable to the towed body, resulting in perturbations both of attitude and position relative to the towing ship. Depending on the sea state and the response of a towing ship, these perturbations can be large enough to throw off the towed body beyond acceptable limits. The maintenance of the attitude of a towed body that is as stable as possible under different towing conditions is one of the major concerns of users.

If an underwater towing system is designed preferably in the early stage, this will save a lot of time and money and shorten the development cycle. The study of hydrodynamic characteristics plays an important role in the preliminary design of underwater towing systems. At present, researchers have carried out a lot of beneficial exploration and research on the hydrodynamic characteristics of underwater towing systems. There are two main methods to study the motion of towing systems. One method is experimental, and the other is numerical calculation. Researchers prefer the experimental method because reliable data and conclusions can be obtained. Bettles and Chapman [1] carried out laboratory experiments on an underwater towing system in a towing tank with the dimension of $9.7 \times 1.8 \times 1.2$ m (l × h × w). In their experiments, the towed body was used as a source electrode connected to a supply, and the trajectory of the towed body was recorded via the electric capacity change between the towed body and fixed horizontal and vertical sensing wires which were positioned at each corner of the cross section and two ends of the towing tank when the towed body was towed at an unsteady velocity. Jiaming Wu carried out a lot of experimental work to study the dynamics of towing systems [2]. He developed an experimental set-up to investigate the hydrodynamic response of a towed body by applying a two-part tow method [3]. The results qualitatively verified the mathematical model of a two-part underwater towing system. Subsequently, Wu [4] proposed an alternative type of controllable underwater towing system and detected its hydrodynamic performances by experimental techniques. To obtain the precious experimental data, Hopland carried out a series of experiments on two kinds of cables with different densities for cable lengths of 300 m and 360 m [5].

The experimental method plays an irreplaceable role in studying the dynamic response of towing systems. However, it is very expensive to carry out experimental studies on towing systems. Generally, experiments are only carried out for towing cables with smaller lengths. Research into ocean towing systems is becoming more and more complex, and with the experimental method, it is difficult to meet the needs of researchers, so numerical calculations came into being, such as the finite element method, the finite difference method, the lumped mass method, etc. Based on the finite element method and the finite difference method, Ablow and Schechter's [6] model can be used to calculate the motion of towing cable systems. Choo and Casarella [7] and Chapman [8] carried out steady state analysis of underwater towing systems. The results of both studies played an important role in the subsequent research on the configuration of towing cables, and the concept of critical radius was given by Chapman in his paper. The research of Kishore and Ganapathy [9] on the steady rotation of towing cables is more comprehensive, in which the function relationship between the turning radius, the length of the cable and the towing speed was given. Koh and Rong [10] carried out an analysis using the cable hydrodynamic model with the finite difference method, in which the cable motion position at each moment is taken as the variable to obtain the approximate governing equation of the cable. Sun [11] used a single-node finite element method to calculate the dynamic model of the cable and the position of each node, which greatly simplified the calculation process of the finite element method. Wang and Sun [12] developed a dynamic model of cables based on Ablow's classical model and overemphasized the numerical algorithm. The dynamic response of a towing cable system to ship maneuver was parametrically simulated with three dimensionless parameters: the ratio of total length to turning radius, ratio of cable mass to vehicle mass and ratio of mass unit length to hydrodynamic force. Yuan [13] developed a newly finite difference method for solving the nonlinear dynamic equations of towing systems. Subsequently, based on the method, he simulated the transient behaviors of a towing system as a towing ship makes 180 and 360 turns with different turning radii and found that the depth and attitude of the towing system were affected by the towing ship turning maneuvers [14]. Based on the finite element, Luis [15] studied the towing maneuver for floating and submerged bodies and the behavior of mooring systems. Zhao [16] also used the finite element method to analyze a fully coupled, three-dimensional dynamic model of a towing cable body system. The numerical results matched the sea trial data

well. According to the literature, numerical methods can be applied to design towing ship turning maneuvers, towing systems and control methods, and the reasonable results show that these numerical models are useful tools with several applications to towing system design, study or optimization.

With the popularity of the computer technique, the numerical calculation method is becoming more and more important in solving complex problems. Among these numerical methods, the lumped mass method is a special form of finite element method, which has been widely used. Huang [17] took a towing cable as a lumped mass model and used the difference scheme to solve it. Based on the previous research results, the lumped mass method was summarized and extended by Chai's paper [18], in which the influence of the seabed interaction on the towing cable was considered, and the numerical simulation was carried out to study the motion characteristics of the cable, riser and towing cable under various conditions. Based on the lumped mass method, Wang [19] dispersed the cable into a series of massless springs and lumped mass points and established the dynamic model of the cable. The dynamic response of the cable with variable length and the dynamic behavior when the cable contacted the seabed was simulated. On the basis of Zhu [20] and Chai, the lumped mass method was applied to the modeling of the multi branch towed linear array by Yang [21], which showed the flexibility of the lumped mass method for the analysis of the multi branch towed line array. Hill [22] studied the hydrodynamic loads and response of a mid-water arch structure using the lumped mass method. The simulation results are in good agreement with the experiment. Du [23] simulated the dynamic behavior of a sonar cable array towed by an underwater vehicle. The strongly nonlinear towed sonar cable array was modeled by the lumped mass method, where the cable array is discretized into a series of massless springs and lumped mass points. Rinaldi [24] developed a modular mooring system and analyzed the dynamic response of the cable using the lumped mass method. Wang [25] proposed a new subsea suspended manifold concept and discussed its feasibility by the hydrodynamic performance state in the lumped mass method. According to the literature, as a special form of finite element method, the lumped mass method is convenient and flexible, and it is easy to add other elements to the boundary of discrete model, such as a buoy, heavy block and so on. Up to now, it has been widely used in the mechanical analysis of ocean towing systems as well as other fields [26,27].

In these analyses, the towed body is modeled as a rigid body with six degrees of freedom, and the hydrodynamic loads acting on the body are represented by a set of dimensionless hydrodynamic coefficients that can be determined experimentally by towing a towed body in a water tank with specified motions. Among these hydrodynamic coefficients, the Munk moment coefficient represents the value of Munk moment which is caused by axial flow. When a three-dimensional towed body moves in a steady straight line at a certain attack angle, the Munk moment is composed of two forces of equal size and opposite direction on the front half and the rear half. The Munk moment seriously affects the motion stability of a towed body. However, looking at the research situation, there are few studies on the influence of Munk moment on the hydrodynamic performance of ocean towing systems. Therefore, it is necessary to study the hydrodynamic response of ocean towing systems under Munk moment. In this paper, the lumped mass method is used to discrete a towing cable into the lumped mass model. At the same time, on the basis of some assumptions, the relationship between the expression of Munk moments in the classical towed body kinematics and the expression of Munk moments in the hydrodynamic analysis software OrcaFlex is established. Based on the above derivation, combined with the specific parameters of a certain sea condition and towing cable system, the hydrodynamic response of a marine cable system under different Munk moment coefficients is modeled by OrcaFlex, and its motion is calculated by the time-domain-coupled dynamic analysis method. In order to ensure the authenticity of the simulation, the simulation time step must be less than the shortest natural node cycle, which should not exceed the shortest natural cycle 1/10. Based on the hydrodynamic performance of the system, some suggestions are given.

## 2. Computational Theory

### 2.1. Mathematical Model

The towing cable model appears as a slender, flexible cylindrical cable. The discrete lumped mass model is used to solve the nonlinear boundary value problem. The basic idea of this model is to divide the towing cable into $N$ segments, and the mass of each element is concentrated on one node, so that there are $N + 1$ nodes. The tension $T$ and shear $V$ acting at the ends of each segment can be considered to be concentrated on a node, and any external hydrodynamic load is considered to be concentrated on the node. The equation of motion of the $i$-th node ($i = 0, 1 \ldots N$) is:

$$M_{A_i}\ddot{R}_i = T_{e_i} - T_{e_{i-1}} + F_{dI_i} + V_i - V_{i-1} + w_i\Delta\bar{s}_i \tag{1}$$

Among them, $R$ represents the node position of the cable.

$M_{Ai} = \Delta\bar{s}_i\left(m_i + \frac{\pi}{4}D_i{}^2(C_{an}-1)\right)I - \Delta\bar{s}_i\frac{\pi}{4}D_i{}^2(C_{an}-1)(\tau_i \otimes \tau_{i-1})$ is the mass matrix of a node, $I$ is the $3 \times 3$ identity matrix; $T_{ei} = EA\varepsilon_i = EA\frac{\Delta s_{0i}}{\Delta s_{\varepsilon i}}$ stands for effective tension at a certain node; $\Delta s_{0i} = \frac{L_0}{(N-1)}$ represents the original length of each segment; $\Delta s_{\varepsilon i} = |R_{i+1}-R_i|$, the stretched length of each segment; EA, axial stiffness of the cable.

$F_{dI_i}$ represents the external hydrodynamics of each node, which is calculated according to the Morison equation:

$$F_{dI_i} = \frac{1}{2}\rho D_i\sqrt{1+\varepsilon_i}\Delta\bar{s}_i(C_{dn_i}|v_{ni}|v_{ni} + \pi C_{dt_i}|v_{ti}|v_{ti}) + \frac{\pi}{4}D_i{}^2\rho C_{an_i}\Delta\bar{s}_i(a_{wi}-(a_{wi}\cdot\tau_i))\tau_i \tag{2}$$

where $\rho$ is the density of sea water, $D_i$ is the diameter of each cable, $C_{dni}$ is the normal drag coefficient, $C_{dti}$ is the tangential drag coefficient and $C_{ani}$ is the inertia coefficient.

In $V_i = \frac{EI_{i+1}\tau_i\times(\tau_i\times\tau_{i+1})}{\Delta s_{\varepsilon i}\Delta s_{\varepsilon i+1}} - \frac{EI_i\tau_i\times(\tau_{i-1}\times\tau_i)}{\Delta s_{\varepsilon i}{}^2} + \frac{H_{i+1}\tau_i\times\tau_{i+1}}{\Delta s_{\varepsilon i}}$, $V$ represents the shear force at the node and $H$ is the torsion.

### 2.2. Expression of Munk Moments

#### 2.2.1. Basic Assumptions

Slender bodies in near-axial flow experience a destabilizing moment called the Munk moment. In order to derive Munk moments, the following assumptions are made:

(1) The towed body is considered as a rigid body;
(2) The shape of the towed body is symmetric with respect to the $xOy$ plane and the $xOz$ plane. In the calculation of the additional mass, the asymmetry of the $xOz$ plane that may be caused by the geometric shape of the towed body is ignored;
(3) The inertial product of the towed body is not taken into account, which means $J_{xy} = J_{yx} = J_{xz} = J_{zx} = J_{yz} = J_{zy} = 0$;
(4) The towed body is completely submerged in the fluid medium and in a fully wet state;
(5) The change of the mass and mass distribution of the towed body in the course of navigation is negligible;
(6) Without considering the Earth's autobiography and the curvature of the Earth, the ground coordinate system is regarded as the inertial coordinate system.

#### 2.2.2. Selection of Coordinate System

In inertial coordinate system $O_0x_0y_0z_0$, $O_0$ is the coordinate origin; $Oxyz$ is the towed body coordinate system and $Ox_1y_1z_1$ is the velocity coordinate system; in both, the coordinate origin is taken at the center of buoyance $O$.

#### 2.2.3. Definition of Kinematic Parameters

(1) Position coordinates and velocity: $r = [x_0, y_0, z_0]^T$, $v = [v_x, v_y, v_z]^T$, $v_0 = [v_{x0}, v_{y0}, v_{z0}]^T$
(2) Attitude angle and angular velocity: $\Omega = [\theta, \psi, \phi]^T$, $\omega = [\omega_x, \omega_y, \omega_z]^T$

(3)    Rudder angle and trajectory angle $\delta = [\delta_e, \delta_r, \delta_d]^T$, $\angle = [\Theta, \Psi, \Phi_c]^T$
(4)    Angle of attack $\alpha$ and sideslip angle $\beta$

Angle of attack $\alpha$: the angle between the projection of the $Ox_1$ axis of the velocity coordinate system in the $xOy$ plane of the towed body coordinate system and the $Ox$ axis of the towed body coordinate system.

Sideslip angle $\beta$: the angle between the $Ox_1$ axis of the velocity coordinate system and the $xOy$ plane of the towed body coordinate system.

2.2.4. Kinetic Equation and Kinematic Equation

The equations of motion of the towed body include kinetic equations and kinematic equations. The kinetic equation of the rigid body is used to describe the relationship between the force, acceleration and angular acceleration of the rigid body; the kinematic equations describe the dynamic relationship between the spatial position and the attitude and velocity and angular velocity.

According to the momentum theorem and the theorem of moment of momentum, the kinetic equation of the towed body can be obtained (the towed body has a small maneuver, the second-order term of the moving parameter of the towed body is ignored and the centroid position is the first order).

$$(m + \lambda_{11})\dot{v}_x = T - C_{xS}\frac{1}{2}\rho v^2 S - \Delta G \sin\theta \tag{3}$$

$$(m + \lambda_{22})\dot{v}_y + (mx_c + \lambda_{26})\dot{\omega}_z + mv_x\omega_z = \frac{1}{2}\rho v^2 S\left(C_y^\alpha \alpha + C_y^{\delta_e}\delta_e + C_y^{\bar{\omega}_z}\bar{\omega}_z\right) - \Delta G \cos\theta\cos\phi \tag{4}$$

$$(m + \lambda_{33})\dot{v}_z - (mx_c + \lambda_{35})\dot{\omega}_y - mv_x\omega_y = \frac{1}{2}\rho v^2 S\left(C_z^\beta \alpha + C_z^{\delta_r}\delta_r + C_z^{\bar{\omega}_y}\bar{\omega}_y\right) + \Delta G \cos\theta\sin\phi \tag{5}$$

$$(J_{xx} + \lambda_{44})\dot{\omega}_x - mv_x\left(y_c\omega_y + z_c\omega_z\right) = \frac{1}{2}\rho v^2 SL\left(m_x^\beta\beta + m_x^{\delta_r}\delta_r + m_x^{\delta_d}\delta_d + m_x^{\bar{\omega}_x}\bar{\omega}_x + m_x^{\bar{\omega}_y}\bar{\omega}_y\right) + G\cos\theta\left(y_c\sin\phi + z_c\cos\phi\right) + \Delta M_{xp} \tag{6}$$

$$(J_{yy} + \lambda_{55})\dot{\omega}_y - (mx_c - \lambda_{35})\dot{v}_z + mx_cv_x\omega_y = \frac{1}{2}\rho v^2 SL\left(m_y^\beta\beta + m_y^{\delta_r}\delta_r + m_y^{\bar{\omega}_x}\bar{\omega}_x + m_y^{\bar{\omega}_y}\bar{\omega}_y\right) - G(x_c\cos\theta\sin\phi + z_c\sin\theta) \tag{7}$$

$$(J_{zz} + \lambda_{66})\dot{\omega}_z + (mx_c + \lambda_{26})\dot{v}_y + mx_cv_x\omega_z = \frac{1}{2}\rho v^2 SL\left(m_z^\alpha\alpha + m_z^{\delta_e}\delta_e + m_z^{\bar{\omega}_z}\bar{\omega}_z\right) + G(y_c\sin\theta - x_c\cos\theta\cos\phi) \tag{8}$$

where $m$ is the mass of towed body; $\lambda$ is the added mass; $\dot{v}_x$, $\dot{v}_y$ and $\dot{v}_z$ are the acceleration of the towed body in m along the towed body coordinate system direction of $x$, $y$ and $z$, respectively; $T$ is the propulsive force; $C_{xS}$ is the drag coefficient of the maximum cross section $S$ as the characteristic area; $\rho$ is the density of sea water; $S$ is the maximum cross-sectional area; $\Delta G = G - B$ is the negative buoyancy of the towed body; $x_c$, $y_c$ and $z_c$ are the centroid position coordinates; $\dot{\omega}_x$, $\dot{\omega}_y$ and $\dot{\omega}_z$, respectively, are the angular accelerations along the $x$, $y$ and $z$ directions of the towed body in the towed body coordinate system; $C_y^\alpha$ and $C_y^{\delta_e}$, respectively, are the derivative of the lift factor of the towed body with respect to the angle of attack $\alpha$ and the derivative of the position with respect to the horizontal rudder angle $\delta_e$; $C_z^\beta$ and $C_z^{\delta_r}$, respectively, are the derivative of the side force factor with respect to the sideslip angle $\beta$ and the derivative with respect to the vertical rudder angle $\delta_r$; $C_y^{\bar{\omega}_z}$ is the dimensionless factor of the lift force for the rotational derivative of the angular velocity $\omega_z$; $C_y^{\bar{\omega}_y}$ is the dimensionless factor of the lateral force for the rotational derivative of the angular velocity $\omega_y$; $\bar{\omega}$ is the dimensionless angular velocity $\omega$; $m_x^\beta$ and $m_y^\beta$, respectively, are the position derivatives of the roll moment factor and yaw moment factor with respect to the sideslip angle $\beta$; $m_z^\alpha$ is the position derivative of the pitching moment factor of a towed body with respect to the attack angle $\alpha$; $m_x^{\delta_r}$ and $m_x^{\delta_d}$, respectively, are the position derivatives of the roll moment factor of a towed body with respect to the vertical rudder angle $\delta_r$ and the differential rudder angle $\delta_d$; $m_y^{\delta_r}$ is the derivative of the yaw moment factor with respect

to the vertical rudder angle $\delta_r$; $m_z^{\delta_e}$ is the derivative of the pitching moment factor to the horizontal rudder angle $\delta_e$; $m_x^{\overline{\omega}_x}$ and $m_x^{\overline{\omega}_y}$, respectively, are the rotational derivatives of the rolling moment factor with respect to $\omega_x$ and $\omega_y$; $m_y^{\overline{\omega}_x}$ and $m_y^{\overline{\omega}_y}$, respectively, are the rotational derivatives of the yaw moment factor with respect to $\omega_x$ and $\omega_y$; $m_z^{\overline{\omega}_z}$ is the rotational derivative of the pitching moment factor with respect to $\omega_z$; $L$ is the length of the towed body.

The kinematic equations describe the dynamic relationship between the spatial position, the attitude, the velocity and the angular velocity. In this paper, the kinematic equations are established according to the coordinate system and the transformation matrix.

$$\dot{\theta} = \omega_y sin\phi + \omega_z cos\phi \tag{9}$$

$$\dot{\psi} = \omega_y sec\theta cos\phi - \omega_z sec\theta sin\phi \tag{10}$$

$$\dot{\phi} = \omega_x - \omega_y tan\theta cos\phi + \omega_z tan\theta sin\phi \tag{11}$$

$$\dot{x}_0 = v_x cos\theta cos\psi + v_y(sin\psi sin\phi - sin\theta cos\psi cos\phi) + v_z(sin\psi cos\phi - sin\theta cos\psi sin\phi) \tag{12}$$

$$\dot{y}_0 = v_x sin\theta + v_y cos\theta cos\phi - v_z cos\theta sin\phi \tag{13}$$

$$\dot{z}_0 = -v_x cos\theta sin\psi + v_y(cos\psi sin\phi + sin\theta sin\psi cos\phi) + v_z(cos\psi cos\phi - sin\theta sin\psi sin\phi) \tag{14}$$

$$v^2 = v_x^2 + v_y^2 + v_z^2 \tag{15}$$

$$\alpha = -arctan\left(\frac{v_y}{v_x}\right) \tag{16}$$

$$\beta = arctan(v_z / \sqrt{v_x^2 + v_y^2}) \tag{17}$$

$$sin\Theta = sin\theta cos\alpha cos\beta - cos\theta cos\phi sin\alpha cos\beta - cos\theta sin\phi sin\beta \tag{18}$$

$$\begin{aligned} sin\psi cos\Theta \ &= sin\psi cos\theta cos\alpha cos\beta + cos\psi sin\phi sin\alpha cos\beta \\ &\quad + sin\psi sin\theta cos\phi sin\alpha sin\beta \\ &\quad - cos\psi cos\phi sin\beta + sin\psi sin\theta sin\phi sin\beta \end{aligned} \tag{19}$$

$$sin\Phi_c cos\Theta = sin\theta cos\alpha sin\beta - cos\theta cos\phi sin\alpha sin\beta + cos\theta sin\phi cos\beta \tag{20}$$

where $\dot{\theta}$ is the pitching angular velocity; $\dot{\psi}$ is the yaw angular velocity; $\dot{\phi}$ is the roll angular velocity; $\dot{x}_0$, $\dot{y}_0$ and $\dot{z}_0$ are the velocities of the towed body in the ground coordinate system along the direction of $x$, $y$ and $z$, respectively.

2.2.5. Munk Moment

By solving the above 17 equations, the whole motion parameters of the space motion of the towed body are obtained, in which the Munk moment is applied to the kinetic equation of the towed body in the form of its dimensionless position derivative of the motion states. The forces of the ideal fluid on the towed body are composed of three parts of linear superposition, which are the ideal fluid forces and moments due to the unsteady motion of the towed body, the constant rotation and the steady motion of the towed body.

$$\begin{cases} M_{\alpha i x} = (\lambda_{22} - \lambda_{33})v_y v_z - \lambda_{32}v_y^2 + \lambda_{23}v_z^2 + (\lambda_{21}v_z - \lambda_{31}v_y)v_x \\ M_{\alpha i y} = (\lambda_{33} - \lambda_{11})v_z v_x - \lambda_{13}v_z^2 + \lambda_{31}v_x^2 + (\lambda_{32}v_x - \lambda_{12}v_z)v_y \\ M_{\alpha i z} = (\lambda_{11} - \lambda_{22})v_x v_y - \lambda_{21}v_x^2 + \lambda_{12}v_y^2 + (\lambda_{13}v_y - \lambda_{23}v_x)v_z \end{cases} \tag{21}$$

With the assumption of this paper, the Munk moment formula can be expressed as follows:

$$\begin{cases} M_{\alpha i x} = (\lambda_{22} - \lambda_{33})v_y v_z \\ M_{\alpha i y} = (\lambda_{33} - \lambda_{11})v_z v_x \\ M_{\alpha i z} = (\lambda_{11} - \lambda_{22})v_x v_y \end{cases} \tag{22}$$

As the angle of attack $\alpha = -arctan\left(\frac{v_y}{v_x}\right)$, the sideslip angle, $\beta = arctan(v_z / \sqrt{v_x^2 + v_y^2})$ assuming that the angle of attack and sideslip angle are small, the Munk moment can be rewritten as the following:

$$[M] = \frac{1}{2}v^2 \begin{bmatrix} sin2\beta & 0 & 0 \\ 0 & sin2\beta & 0 \\ 0 & 0 & sin2\alpha \end{bmatrix} \cdot \begin{bmatrix} \lambda_{22} - \lambda_{33} & 0 & 0 \\ 0 & \lambda_{33} - \lambda_{11} & 0 \\ 0 & 0 & \lambda_{11} - \lambda_{22} \end{bmatrix} \quad (23)$$

From the above deduction, the Munk moment is actually proportional to the $v^2$, $sin(2\alpha)$ and $sin(2\beta)$ and $\lambda_{MM}$, which is consistent with the expression of the Munk moment $M = \frac{1}{2} \cdot C_{mm} \cdot M \cdot sin(2\alpha) \cdot v^2$ given in the commercial software OrcaFlex. Here, $\lambda_{MM}$ is the $M \cdot C_{mm}$, the product of the Munk moment coefficient $C_{mm}$ of the OrcaFlex and the mass of the water currently displaced by the towed body $M$. Therefore, in this paper, OrcaFlex is used as the simulation platform to establish the model of the towed cable array under different Munk moment coefficients during direct sailing, and the dynamic response of the towed body is analyzed.

## 3. Numerical Model Setup

The simulated flow field in this paper is a still water environment, neglecting the action of wind, wave and current. The initial state of cable is horizontal. The diameter of cable is 0.025 m. The axial stiffness of the cable is $EA$ = 6000 $N$; the bending stiffness, $EI$ = 0; the mass per unit length, $m$ = 0.0011 t/m. Additionally, the length of the towing cable is 450 m.

The towed body is modeled by the 6D spar buoy. The basic parameters are shown in Tables 1 and 2.

**Table 1.** The basic parameters of the towed body.

| Mass (t) | Mass Moments of Inertia (t.m$^2$) | | | Total Length (m) | Centre of Mass (m) | | |
|---|---|---|---|---|---|---|---|
| | $I_x$ | $I_y$ | $I_z$ | | $x$ | $y$ | $z$ |
| 1.5 | 0.1 | 5 | 5 | 3.9 | 0 | 0 | 0 |

**Table 2.** The basic parameters of the towed body structure.

| Cylinder Segment | ID (m) | OD (m) | Length (m) | Cumulative Length (m) |
|---|---|---|---|---|
| 1 | 0 | 0.2 | 0.1 | 0.1 |
| 2 | 0 | 0.35 | 0.3 | 0.4 |
| 3 | 0 | 0.5 | 1.0 | 1.4 |
| 4 | 0 | 0.5 | 1.0 | 2.4 |
| 5 | 0 | 0.4 | 1.0 | 3.4 |
| 6 | 0 | 0.2 | 0.5 | 3.5 |

Each vessel in the model must specify its own *RAO* in OrcaFlex.

The vessel data, displacement *RAOs*, wave load *RAOs*, wave drift *QTFs*, stiffness, added mass and damping data all come from an *NMIWave* diffraction analysis of a 103 m long tanker in 400 m water depth. The tanker used in this analysis had the following properties: *Breadth* 15.95 m, *Draught* 6.66 m, Transverse *GM* 1.84 m, Longitudinal *GM* 114 m and Block Coefficient 0.804. The diffraction analysis used 8% extra damping in roll about *CG*.

The main sea boundary conditions for the system are the following: the depth of the sea, $D$ = 100 m; the current speed is taken as 0 m/s. To simplify the model, the wave height $H$ is taken as 0.

The vessel speed is 5 m/s; the total simulation time is 210 s. Completed in OrcaFlex, the model is shown in Figure 1.

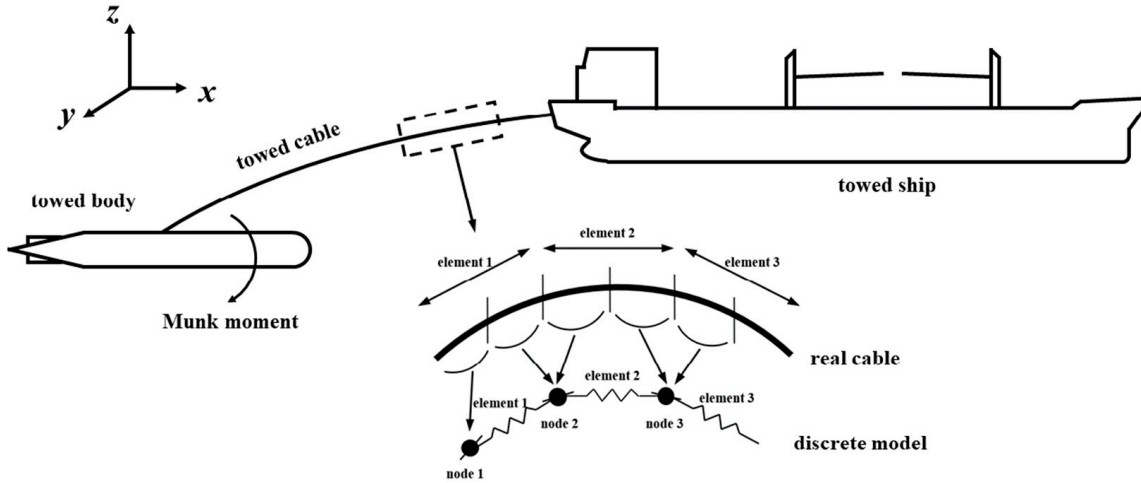

**Figure 1.** Schematic diagram of the ocean towing cable-array system model.

## 4. Results and Discussion

### 4.1. The Variation of the Effective Tension of the Towing Cable

The maximum effective tension of the towing cable along the length direction in Figure 2 shows that the amplitude of the maximum effective tension under the action of a certain single Munk moment coefficient has a tendency of recurrent fluctuations along the length direction: the maximum tension along the direction of the cable length decreases significantly near the zone of the top end (0–70 m), then increases slowly at the distance of 70–100 m along the length direction from the top end, then decreases slowly again; eventually, it presents an increasing trend at the region of 370 m—the bottom end; except for 0.1 and 1, when the Munk moment coefficient is in the other numerical range. With the increase in the Munk moment coefficient, the maximum tension also increases, that is to say, when the Munk moment coefficients are 1 and 0.1, the distribution of the maximum effective tension has a jump and discontinuity in numerical value; when the Munk moment coefficient is 1, the maximum tension value is less than that when the Munk moment coefficient is 0.2, but it is larger than that when the Munk moment coefficient is 0; when the Munk moment coefficient is 0.1, the maximum tension value is greater than that when the Munk moment coefficient is 0.9, but less than that when the Munk moment coefficient is 1.1. The reasons for this phenomenon is that when the Munk moment coefficient is small, the drag by Munk moment makes the underwater towed body rotate, and the flow facing surface area changes constantly. Sometimes this is a large change, and sometimes it is small; when it is large, the drag force and the damping force on the towed body of the current become larger, which makes the towing cable tension larger; this phenomenon is obvious at 0.1 and 1.

When the Munk moment coefficient continues to increase, as a whole, the spatial attitude of the underwater towed body changes more rapidly and sharply. The rotation amplitude and the translation amplitude of the towed body increase, and as a result, the traction effect on the towing cable is more obvious, thus, in the large coefficient range, the maximum tension increases with the increase in the Munk moment coefficient. To observe the changes of minimum effective tension of the towing cable along the length direction, we find that with the comparison of the maximum tension, the minimum distribution along the length direction of the cable tension is relatively single, there is no obvious increase or decrease in cycle repeated tension amplitude along the length direction, and the minimum effective tension along the length direction shows a linear decreasing trend from the top end to the bottom end of the towing cable. The mean effective tension distribution and

the minimum effective tension distribution along the length direction have a similar trend; both of them show the a decreasing trend in the tension value from the top end to the bottom end of the towing cable, and there is little difference between the numerical value of the minimum effective tension and the mean effective tension under the action of a certain Munk moment coefficient. The difference in standard deviation of the effective tension of the towing cable between the different parts of the towed body varies more greatly along the length direction, and the synchronization of each part of the towed body is worse during towing; the smaller the difference in the standard deviation of the effective tension of the towing cable between the different parts of the towed body is, the better the synchronization of each part of the towed body, and the more balanced the force and the more synchronous the effective tension will be. This will undoubtedly improve the stability, coordination and integrity of the whole towing system.

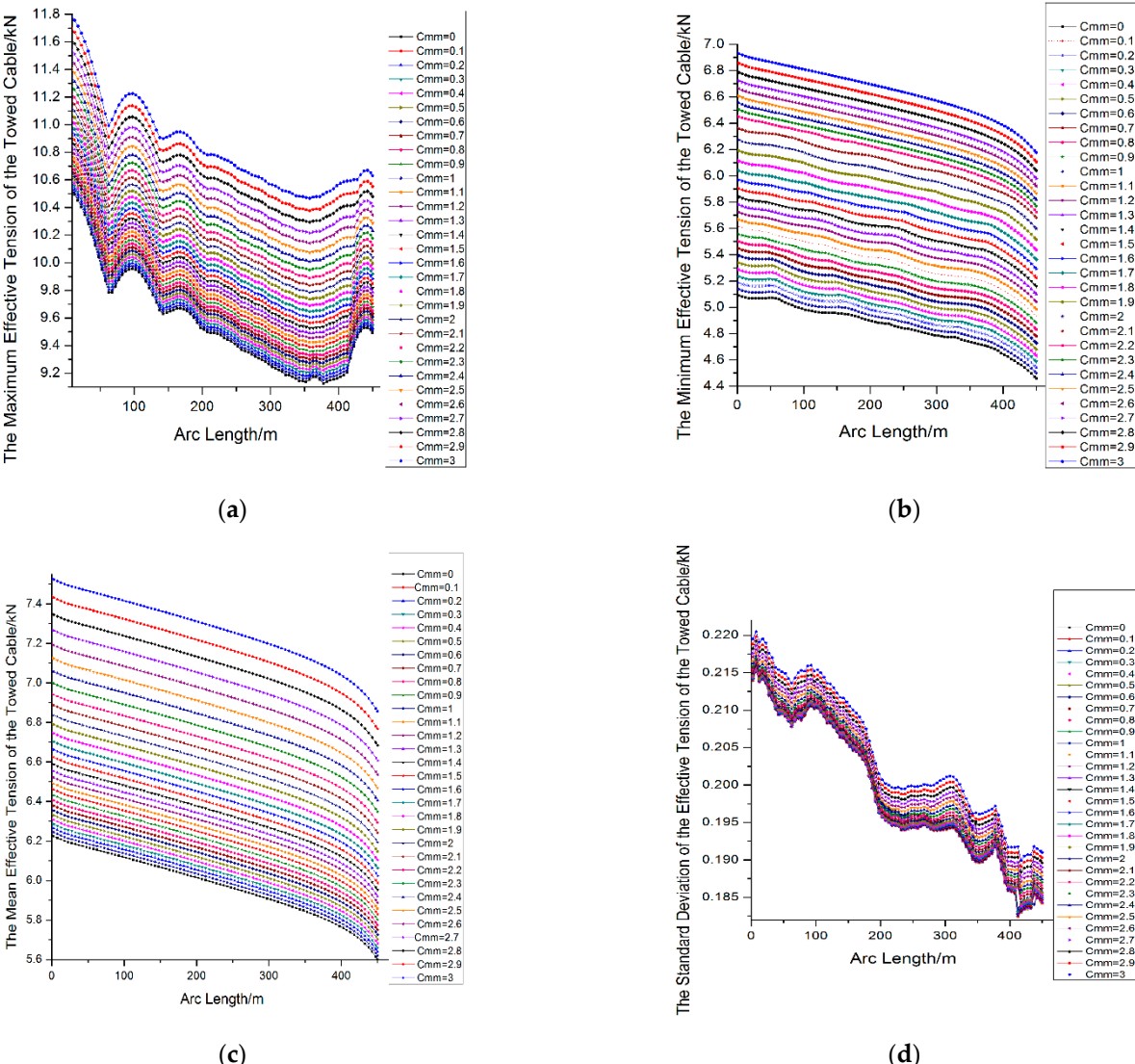

**Figure 2.** (**a**) The maximum effective tension of the towing cable along the length direction; (**b**) the minimum effective tension of the towing cable along the length direction; (**c**) the mean effective tension of the towing cable along the length direction; (**d**) the standard deviation of the effective tension of the towing cable along the length direction.

With the observation of the standard deviation of the effective tension of the towing cable along the length direction, we find that with the increase in the Munk moment coefficient, the standard deviation of cable tension appears to decrease firstly and then

increase in the whole range of the Munk moment coefficient: in the low Munk moment coefficient range (0–0.5), the standard deviation of the effective tension of the towing cable decreases as the Munk moment coefficient increases, and when the Munk moment coefficients are 0 and 0.2, the jump phenomenon of the numerical value of the standard deviation of the effective tension of the towing cable occurs at a distance of 350 m and 345 m from the top end, respectively. When the Munk moment coefficient is 0.5–0.9, the curves of the standard deviation of the cable tension have a high spatial configuration along the length direction, which shows that the effective tension of the towing cable has almost the same synchronism along the length direction. When the Munk moment coefficient increases to 1, the standard deviation of the effective tension of the towing cable begins to increase obviously, and there is an obvious jump phenomenon of the standard deviation at the distance of 345 m from the top end. When the Munk moment coefficient is in the range of 1–1.5, the standard deviation of the effective tension of the towing cable along the length direction is the maximum when the Munk moment coefficient is 1. When the Munk moment coefficient is in the range of 1.1–1.5, the distributions of the standard deviation of the effective tension of the towing cable along the length direction are basically the same. When the Munk moment coefficient is 1.6–3, the standard deviation of the effective tension of the towing cable also increases with the increase in the Munk moment coefficient. When the Munk moment coefficient is 2.2, the standard deviation of the effective tension of the towing cable begins to exceed the standard deviation of the effective tension of the cable when the Munk moment coefficient is 1. When the Munk moment coefficient is 2.3, the standard deviation of the effective tension of the towing cable starts to exceed the standard deviation of the effective tension of the towing cable when the Munk coefficient is 0, except that the standard deviation of the effective tension increases at a small amplitude near the area of the bottom end, and the standard deviation of the effective tension under the action of a certain single Munk moment coefficient decreases successively from the top end to the bottom end along the length direction, which indicates that the effect of the traction of the vessel on the degree of abrupt change of the effective tension is greater than that of towed body. The reason for the jump phenomena of the standard deviation of the effective tension of some Munk moment coefficients is that the motions and attitudes of the towed body are more complicated under the these Munk moment coefficients; therefore, the effect on the configuration of the towing cable is greater than that under other coefficients, and as a result, the tension of the towing cable is more abrupt at some specific positions (345 m, 350 m), where the tension synchronization is obvious different from other zones.

As a whole, the standard deviation of the effective tension of the towing cable has three local extremums when the Munk moment coefficient is in the range of −3; these values appear when the coefficients are 0, 1 and 3, respectively; furthermore, the standard deviation of the effective tension of $C_{mm} = 3>$ the standard deviation of the effective tension of $C_{mm} = 0>$ the standard deviation of the effective tension of $C_{mm} = 1$. That is to say: the coordination and synchronization of the effective tension of the towing cable is not better when there is no Munk moment; the coordination and synchronization of the effective tension of the towing cable may be better than that when there is no Munk moment.

### 4.2. The Variation of Bending of the Towing Cable

Figure 3 shows the variation of bending of the towing cable. By comparing the observation of the distributions of the bending moment and the curvature of the towing cable along the length direction, we find that for a certain wave directions, the curve of the bending moment is a segmented horizontal line parallel to the horizontal direction of the cable length, which has different discontinuous curvature values, and its distribution trend presents the characteristic of ladder distribution; the distributions of the curvature along the length under different wave directions are different from the distribution of the bending moment of the towing cable along the length direction. The distribution of bending moment shows the trend of ladder increase or decrease along the length direction, but the distribution curves of curvature curves show a certain continuity along the length

direction, which indicates that the change of the bending moment has some lag compared with the change of the curvature along the length. This shows that the change of the bending moment along the length direction is not synchronized with the change of the curvature along the length direction; all of those are related to the material quality of the towing cable itself and the damping of the water.

At the same time, the variations of the curvature and the bending moment of the towed body near the towed body are more rapid, which is related to the motions of translation and rotation of the towed body. In the middle region of the towing cable, which is far from the underwater towed body and the top end connected with the vessel, both the bending moment and the curvature are close to 0; the bending moment and curvature of the towing cable at an approximate distance of 20 m from the top end and the bottom end show a large range of fluctuation, and the fluctuation range at the distance of 20 m from the top end is larger than that of the distance of 20 m from the bottom end, which shows that the effect of the motion of the towed body on the bending moment and curvature of the towing cable is greater than that of the vessel motion on the bending moment and curvature of the towing cable.

By comparing the difference between the standard deviation of the bending moment and the standard deviation of the curvature along the length direction, it is found that the moment distribution is not continuous along the cable direction, but the degree of the variation of the bending moment is consistent with the coordination of the change of the bending moment along the cable length (except near the bottom area), and the standard deviation of the bending moment is 0 except the bottom end. The curvature distribution is exactly the opposite, although the curvature distribution along the length direction is continuous, but the coordination degree of the sharp curvature change and the change of curvature along the cable length direction is inconsistent, which can be seen from the ladder distribution of the standard deviation of the curvature along the length direction. That is to say, under different Munk moment coefficients, the bending moment exhibits a ladder distribution along the cable length direction, but the change of the bending moment along the cable length shows better coordination and synchronization; the curvature exhibits better continuity along the cable length, but the standard deviation of the curvature change along the cable length direction shows a ladder-type distribution, which indicates its coordination and synchronization of the curvature along the cable length is poor.

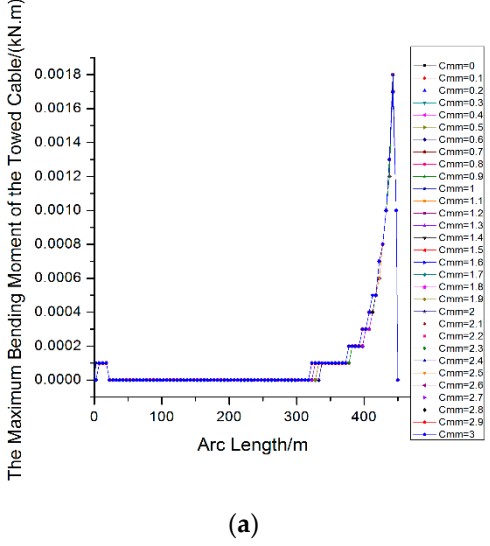

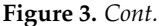

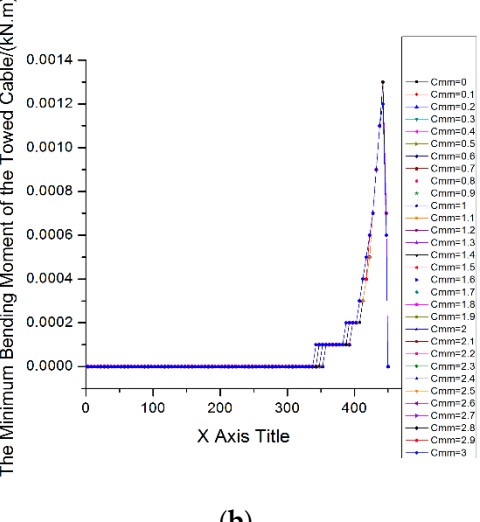

(**a**)                                                          (**b**)

**Figure 3.** *Cont.*

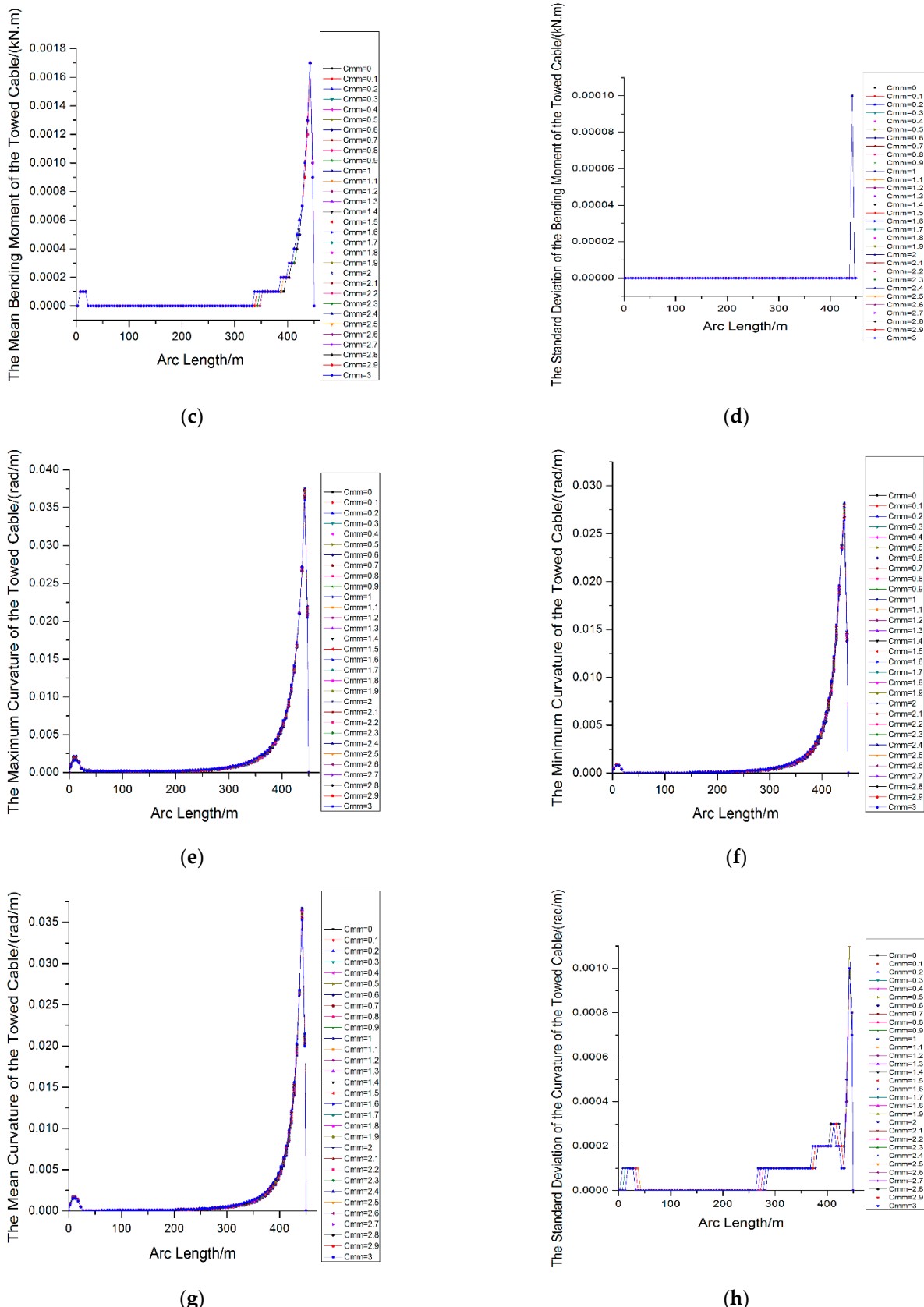

**Figure 3.** (**a**) The maximum bending moment of the towing cable along the length direction; (**b**) the minimum bending moment of the towing cable along the length direction; (**c**) the mean bending moment of the towing cable along the length direction; (**d**) the standard deviation of the bending moment of the towing cable along the length direction; (**e**) the maximum curvature of the towing cable

along the length direction; (**f**) the minimum curvature of the towing cable along the length direction; (**g**) the mean curvature of the towing cable along the length direction; (**h**) the standard deviation of the curvature of the towing cable along the length direction.

*4.3. The Variation of the Towed Body*

Figure 4 shows the dynamic response of the towed body. With the observation of the distribution of the pitching angle in the time domain, it is found that the pitching angle of the towed body presents two trends with the increase in the Munk moment coefficient: first, with the increase in the Munk moment coefficient, the pitching angle of the towed body in the stable state increases successively; second, with the increase in the Munk coefficient, the time turning point of the pitching angle of the towed body which starts from the chaotic state and into the steady state is advanced in proper sequence; before the time turning point, the pitching angle of the towed body keeps decreasing with the increase in the Munk moment coefficient; after the time turning point, the pitching angle of the towed body increases with the increase in the Munk moment coefficient; in the time domain, the pitching angle of the towed body under different Munk moment coefficients eventually approaches a certain stable value, respectively. The reason for the above phenomena is that the pitching and roll of the towed body interact with each other.

It is found that the time turning point of the roll angle into the steady state is in advance of the increase in the Munk moment coefficient in proper sequence; with the further observation of the roll angular curves in the time domain, under different Munk moment coefficients, the roll angle of the towed body is basically maintained at three values when the towing attitude of the towed body is basically stable, which are $360°(C_{mm} = 0,$ 0.7, 1, 1.5, 2.4, 2.9), $0°(C_{mm} = 0.1, 0.4, 0.5, 0.6, 1.1, 1.3, 1.4, 1.8, 2.1, 2.3, 2.5, 2.6, 2.7, 2.8, 3)$ and $-360°(C_{mm} = 0.2, 0.3, 0.8, 0.9, 1.2, 1.6, 1.7, 1.9, 2, 2.2)$. That is to say: when there is no Munk moment ($C_{mm} = 0$), the towed body in the steady state will have a $360°$ roll; when it is under the action of the Munk moments, with the change of the Munk moment coefficient, the roll angle is also different; the roll angle is 0 under some specific Munk moment coefficients, and the towed body will have a $-360°$ roll under some other Munk moments. Compared with the influence of the Munk moment coefficient on the pitching angle, the influence of the Munk moment coefficient on the roll angle of the towed body in the steady state is discrete, and its stable roll angle does not show a single increase or decrease with the Munk moment coefficients.

By comparing the spectral density of the pitching angle and roll angle, it is found that the peak of the density of the pitching angle of the towed body is much smaller than the peak of the spectral density of the roll angle of the towed body in the low frequency range, which indicates that in the low frequency range, the low-frequency response of the roll angle is much larger than the low-frequency response of the pitching angle; compared with the spectral density of roll angle, the spectral density curves of the pitching angle decreases more slowly with the increase in frequency, the spectral density of the pitch angle is basically reduced to 0 at 0.1 Hz; the spectral density curve of the roll angle has some cuspidal points in the process of descending and the curve is not very smooth; the spectral density curve is basically reduced to 0 at 0.05 Hz.

It is found that the yaw angle of the towed body is very small ($-0.0005°$–$0.0004°$) over the whole time domain, indicating that the change of the Munk moment has little effect on the yaw angle and it needs to take more time to reach the steady state under different Munk moment coefficients, which is more than the time required for the roll angle and less than the time required for the pitching angle; the yaw angle of the towed body is basically maintained at two values when the towing attitude of the towed body is basically stable, which are $-0.0001°(C_{mm} = 0, 0.1, 0.2, 0.3, 0.4, 0.5, 0.6, 0.7, 0.8, 0.9, 1, 1.1, 1.2, 1.3, 1.4, 1.5, 1.8,$ 1.9, 2, 2.1, 2.2, 2.8, 2.9) and $0.0001°(C_{mm} = 1.6, 1.7, 2.3, 2.4, 2.5, 2.6, 2.7, 3)$; it is found that there are three yaw angular mutations with large amplitudes in the time domain before the yaw angle of the towed body is stable, which is due to the change of the pitch angle of the towed body. Since the pitching angle of the towed body has not yet entered a steady state,

the pitching process of the towed body will have an impact on the yaw angle, and as the pitch angle reaches a steady state, the yaw angle also gradually reaches a certain steady state. The spectral density of the yaw angle of the towed body is 0, which shows that the yaw angle of the towed body will not fluctuate with the change of the external frequency in the frequency domain, and no low-frequency response will occur. With the increase in the Munk moment coefficient, the water depth after the stability of the towed body increases continuously, and the maximum water depth is about 5 m different from the minimum water depth.

Observing the depth spectral density curves of Z of the towed body, it is found that the spectral density curves of the towed body depth coincide with each other under different Munk moment coefficients, and all the spectral density curves change to 0 at 0.1 Hz; this indicates that the change of the heave of the towed body is not sensitive to the change of frequency. With the observation of Y of the towed body versus time, it is found that the lateral (sway) displacement of the towed body is very small under the action of different Munk moment coefficients, and the lateral (sway) displacement after the stabilization of the towed body is basically symmetrical on both sides of Y = 0; when there is no Munk moment, the lateral displacement of the towed body gradually tends to be stable after 100 s, and the lateral (sway) displacement of the towed body of 180 s reaches a steady state, and finally, it turns towards the positive side of Y; when the Munk moment coefficient is taken into account, as the Munk coefficient increases, the time to reach the lateral (sway) steady state is advanced in turn, and before its stabilization, Y will have a ladder distribution increase trend in the time domain until the towed body lateral displacement reaches a certain steady state. It is found that all the time domain curves of X of the towed body are basically coincident with other different Munk moment coefficients; the curve is an inclined line with a slope of the size of the vessel speed, which shows that the towing speed of the vessel has the main influence on the motion of the X direction of the towed body, and the change of the Munk moment has a very weak influence on the dynamic response of the X of the towed body; it is found that the spectral density curves along the X-direction of the M-moments are reduced to 0 at 0.1 Hz, which indicates that the dynamic response of the towed body along the X-direction is also concentrated in the low-frequency region, and the change of the Munk moment coefficient has little effect on the spectral density of the towed body. From the order of magnitude, the influence of the Munk moment on the heave direction of the towed body is much larger than that of the sway and surge directions, which indicates that the change of the Munk moment mainly affects the heave response of the towed body. The influence of the Munk moment on the sway and yaw response of the towed body is very weak, but the influence on the sway direction is greater than that of the surge direction.

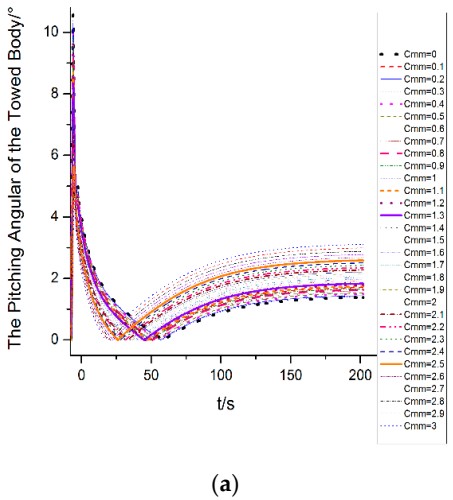

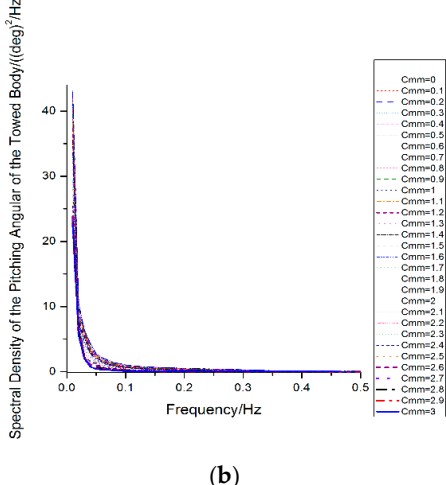

(a)

(b)

**Figure 4.** *Cont.*

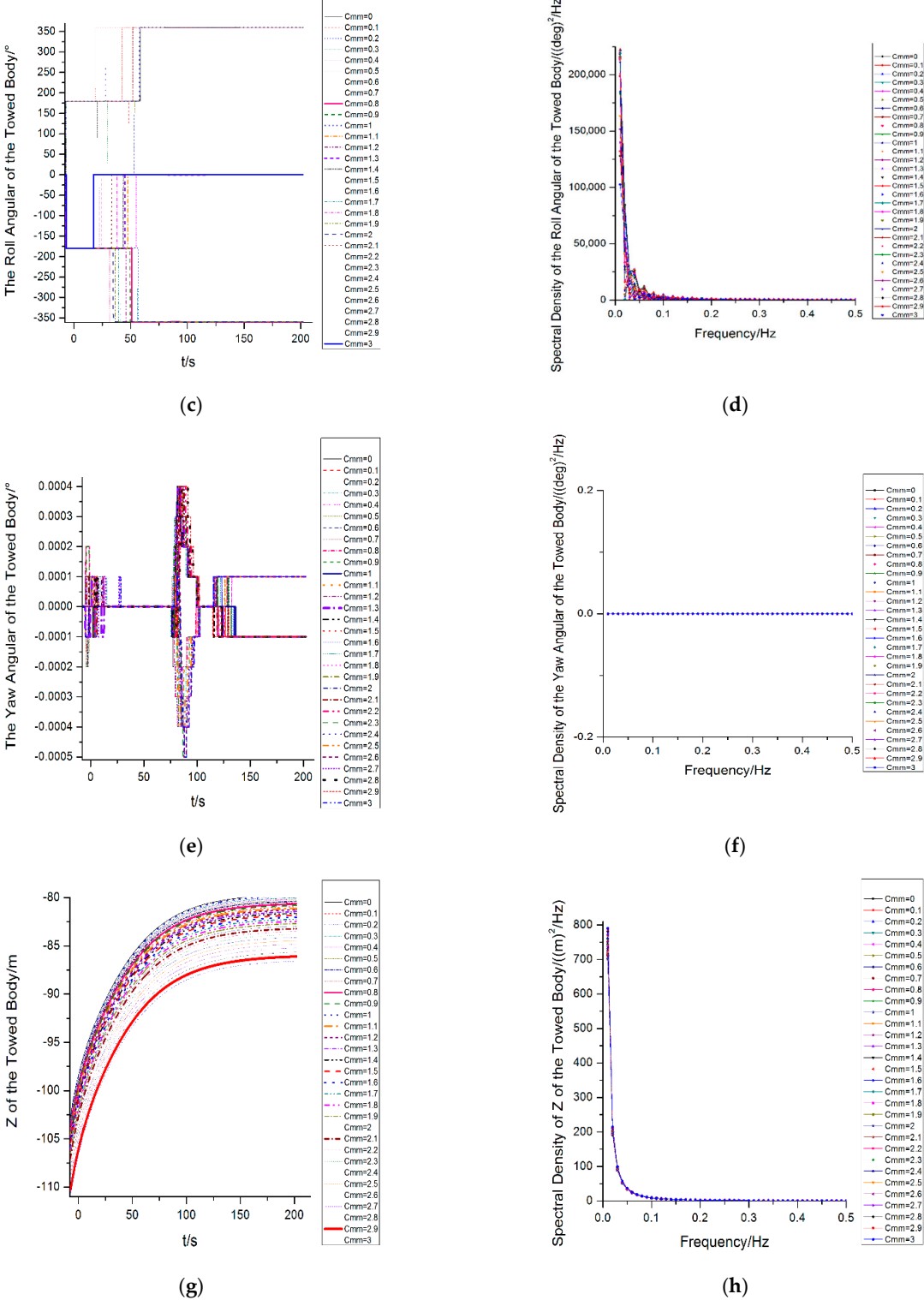

**Figure 4.** *Cont.*

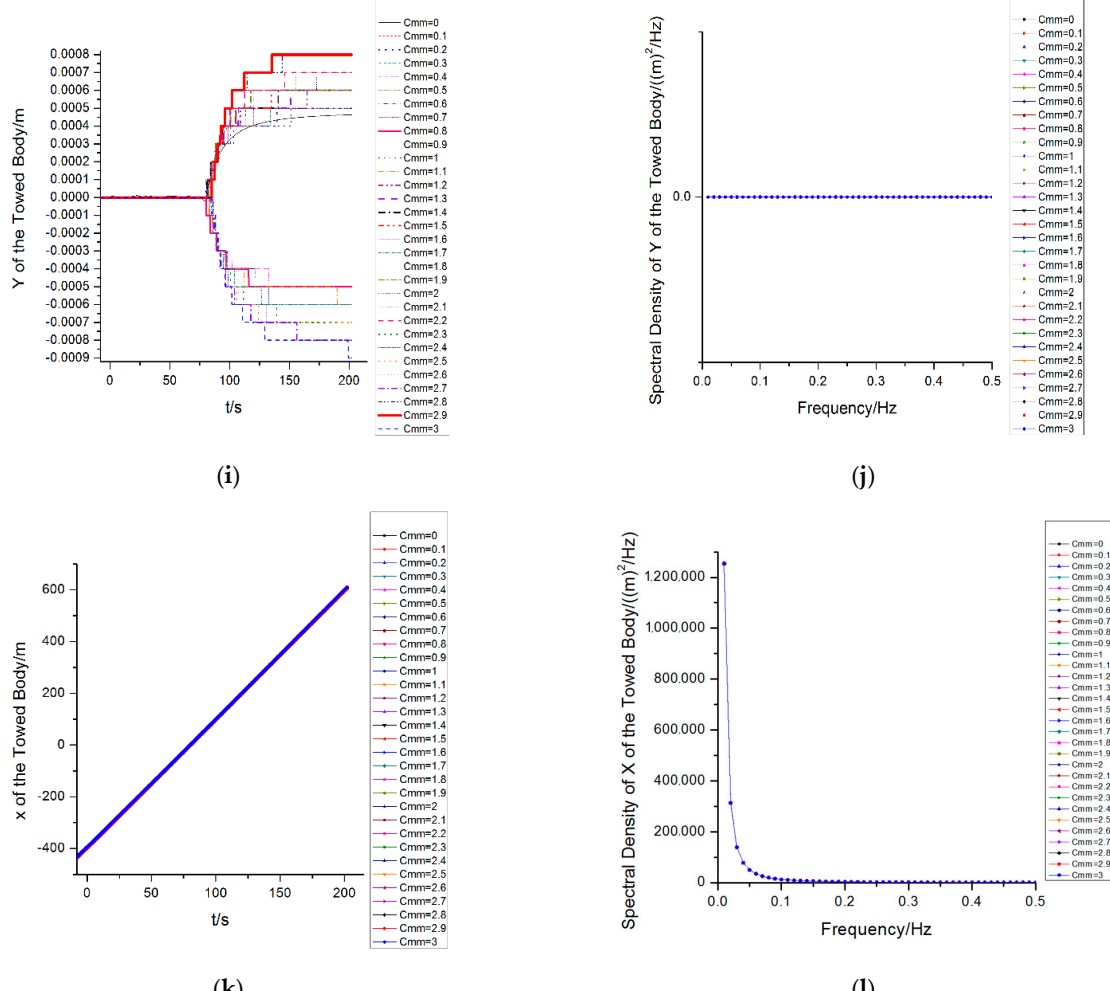

**Figure 4.** (**a**) Pitching angle of the towed body versus time; (**b**) spectral density of the pitching angle of the towed body; (**c**) roll angle of the towed body versus time; (**d**) spectral density of the roll angle of the towed body; (**e**) yaw angle of the towed body versus time; (**f**) spectral density of the yaw angle of the towed body; (**g**) Z of the towed body versus time; (**h**) spectral density of Z of the towed body; (**i**) Y of the towed body versus time; (**j**) spectral density of Y of the towed body; (**k**) X of the towed body versus time; (**l**) spectral density of X of the towed body.

## 5. Conclusions

In this paper, using the lumped mass method, the hydrodynamic response of an ocean towing cable-array system under different Munk moment coefficients was studied. This study demonstrated the following:

(1) The amplitude of the maximum effective tension under the action of a certain single Munk moment coefficient has a tendency of recurrent fluctuations along the length direction. The coordination and synchronization of the effective tension of the towing cable may be better than that when there is no Munk moment; when the Munk moment coefficient is 0.5–0.9, the effective tension of the towing cable has almost the same synchronism along the length direction; with the increase in the Munk moment coefficients, the synchronism of the tension of the towing cable firstly decrease and then increase.

(2) When the Munk moment coefficient is small, the drag by the Munk moment makes the underwater towed body rotate, and the flow facing surface area changes constantly. Sometimes, this is a large change, and sometimes it is small; when it is large, the drag

force and the damping force on the towed body of the current become larger, which makes the towing cable tension larger.

(3) Under different Munk moment coefficients, the bending moment exhibits a poor continuity along the cable length direction, but the change of the bending moment along the cable length shows better coordination and synchronization; the curvature exhibits better continuity along the cable length, but its coordination and synchronization of the curvature along the cable length is poor.

(4) With the increase in the Munk moment coefficient, the pitching angle of the towed body in the stable state increases successively and eventually approaches a certain stable value with the increase in the Munk coefficient. The pitching and roll of the towed body interact with each other; the time turning point of the roll angle into the steady state is in advance of the increase in the Munk moment coefficient in proper sequence; under different Munk moment coefficients, the roll angle of the towed body is basically maintained at three values when the towing attitude of the towed body is basically stable. The Munk moment has little effect on the yaw angle, the yaw angle of the towed body will not fluctuate with the change of the external frequency in the frequency domain, and no low-frequency response will occur.

(5) With the increase in the Munk moment coefficients, the water depth after the stability of the towed body increases continuously, and the maximum water depth is about 5 m different from the minimum water depth; the change of the heave of the towed body is not sensitive to the change of frequency. The time taken to reach the lateral (sway) steady state is advanced in turn and before its stabilization. The change of the Munk moment has a very weak influence on the dynamic response of the X-direction of the towed body

(6) From the order of magnitude, the influence of the Munk moment on the heave direction of the towed body is much larger than that of the sway and surge directions, which indicates that the change of the Munk moment mainly affects the heave response of the towed body. The influence of the Munk moment on the sway and yaw response of the towed body is very weak, but the influence on the sway direction is greater than that of the surge direction.

(7) The study enriched the research on the dynamic characteristics of ocean towing systems and helps to design a more stable towing system, which will play a more efficient role in the exploration of marine resources and sustainability in ocean engineering.

**Author Contributions:** Conceptualization, D.Z. and B.Z.; methodology, K.Z.; software, D.Z.; validation, D.Z., K.Z.; investigation, D.Z.; resources, D.Z.; data curation, D.Z.; writing—original draft preparation, D.Z.; writing—review and editing, B.Z.; visualization, B.Z. All authors have read and agreed to the published version of the manuscript.

**Funding:** This research was funded by Program for Scientific Research Start-up Funds of Guangdong Ocean University, grant number 060302072101.

**Institutional Review Board Statement:** Not applicable.

**Informed Consent Statement:** Not applicable.

**Data Availability Statement:** Not applicable.

**Conflicts of Interest:** The authors declare no conflict of interest.

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
