# Peer review of "Hydrodynamic Response of Ocean-Towed Cable-Array System under Different Munk Moment Coefficients"

_sustainability, doi:10.3390/su14031932_

Round 1
Reviewer 1 Report
Generally.
I found that this paper plays an important role in the operation of the ocean in the ocean towing system. There is a lot of useful information in this paper, and I could deepen my understanding. This paper shows that the form of motion of the towing body is closely linked to the motion of the towing cable, and the motion state is expressed by discretizing the towing cable into a lumped mass model using the lumped mass method. The results show that the expression of Munk moments in classical towing body kinetics can predict complex three-dimensional motions and provide a theoretical basis for the optimal design of cables and towed bodies. It seems to me that this is useful information. However, there are many theories that are inadequate for use in design, and it would be better to revise the introduction as basic research.
Sentence
- L260 Please explain the reason why you have chosen to model it this way. This case seems to depend on the geometry, so why don't you show a simplified modeling and the problem you want to solve?
- In Fig.2, there seems to be a correlation in each coefficient. I think it would be easier to understand and read the characteristics if you focus on a specific case.
- Fig.4 I did not understand the interpretation of this diagram. I think it shows the effect of angle, but I think you should narrow down the number of cases and describe it.
4.I feel that the lumped mass method is like the mechanics of a quality point system. What is your interpretation of doing this while neglecting shapes that are affected by mass and moment of inertia? Also, is the Munk moment likely to change depending on the dynamical system?
- L535 ‘when the Munk moment coefficient is 0.5-0.9, the effective tension of the towed cable has almost the same synchronism along the length direction; with the increase of the Munk moment coefficients, the synchronism of the tension of the towed cable decrease firstly and then increase.’ I think that it seems to me that there is some mechanical meaning to this coefficient, but how is it interpreted?
L552 The effect of the motion of the towing body on the bending moment and curvature of the towing cable is greater than the effect of the motion of the ship on the bending moment and curvature of the towing cable. This sentence is normally interpretation. Is it worth mentioning?
L596 I have some doubts about the evaluation of the spectral density curve. First of all, the boundary between the low frequency region and the high frequency region is not clear, and it seems to be somewhat different from the concept of impact loading. First of all, it would be better to clarify the process of filtering, so that it can be classified by the effect of frequency.
Reviewer 2 Report
Dear Authors,
My only concern is about the conclusions. It is too lengthy and may be split into two sections as discussions and conclusions
1. The article is an interesting subject in ocean engineering especially in piping and towing sector.
2. The authors had an indepth research on the literature.
3. The manuscript applies the theory and implementation of Munk Moments to the problem sucessfully.
4. They set up a numerical scheme for the validation of the theory.
5. The paper is quite organized and well written.
6. The paper will contribute to the field to a degree.
7. The conclusions are consistent with the findings, however it may have been better organized.
